# Correlation between Collagen Type I/III Ratio and Scar Formation in Patients Undergoing Immediate Reconstruction with the Round Block Technique after Breast-Conserving Surgery

**DOI:** 10.3390/biomedicines11041089

**Published:** 2023-04-04

**Authors:** Hyo-young Kim, Ho-young Im, Hee-kyung Chang, Hwan-do Jeong, Jin-hyung Park, Hong-il Kim, Hyung-suk Yi, Yoon-soo Kim

**Affiliations:** 1We Are the Plastic Surgery, 415, Haeun-daero, Haeundae-gu, Busan 48064, Republic of Korea; 2Department of Plastic and Reconstructive Surgery, College of Medicine, Kosin University, 262, Seo-gu, Busan 49267, Republic of Korea; 3Department of Pathology, College of Medicine, Kosin University, 262, Seo-gu, Busan 49267, Republic of Korea; 4Kosin Innovative Smart Healthcare Research Center, Kosin University Gospel Hospital, 262, Seo-gu, Busan 49267, Republic of Korea

**Keywords:** breast-conserving surgery, collagen type I, collagen type III, Vancouver Scar Scale

## Abstract

The aim of this study was to investigate the relationship between the collagen type I/III ratio and scarring in patients who underwent immediate reconstruction with the round block technique (RBT) after breast conservation surgery. Seventy-eight patients were included, and demographic and clinical characteristics were recorded. The collagen type I/III ratio was measured using immunofluorescence staining and digital imaging, and scarring was assessed using the Vancouver Scar Scale (VSS). The mean VSS scores were 1.92 ± 2.01 and 1.79 ± 1.89, as assessed by two independent plastic surgeons, with good reliability of the scores. A significant positive correlation was found between VSS and the collagen type I/III ratio (r = 0.552, *p* < 0.01), and a significant negative correlation was found between VSS and the collagen type III content (r = −0.326, *p* < 0.05). Multiple linear regression analysis showed that the collagen type I/III ratio had a significant positive effect on VSS (β = 0.415, *p* = 0.028), whereas the collagen type I and collagen type III content had no significant effect on VSS. These findings suggest that the collagen type I/III ratio is associated with scar development in patients undergoing RBT after breast conservation surgery. Further research is needed to develop a patient-specific scar prediction model based on genetic factors affecting the collagen type I/III ratio.

## 1. Introduction

Post-operative scarring can cause functional impairment as well as cosmetic problems, negatively affecting a patient’s quality of life [1].

Pathological scars can take the form of elongated, depressed, hypertrophic and keloid scars. This can result in symptoms such as itching or pain at surgical sites.

Scarring is a complex process influenced by many factors, including the mechanical forces applied to the skin. In areas where the skin is stretched or tensioned, the resulting scars may become more pronounced or develop hypertrophic or keloid characteristics [2]. In addition to collagen content and ratios, the effects of mechanical forces on scar formation and healing must be considered. Excessive mechanical forces can stimulate fibroblast proliferation, collagen synthesis and inflammation, leading to hypertrophic or keloid scars.

Strategies to minimize these forces and improve scar quality can be employed in the immediate postoperative period, such as the use of silicone-based creams, compression garments and appropriate wound care techniques [3,4]. Addressing these factors in conjunction with assessing collagen levels may provide a more comprehensive approach to predicting and improving scar outcomes in breast cancer patients undergoing breast-conserving surgery. Steri-strip^TM^ can help reduce mechanical forces by providing a stable closure of the wound edges and preventing them from spreading or pulling apart. Steri-strip^TM^ can also create a moist environment that reduces friction and shear stress on the wound surface [5]. By reducing mechanical forces, Steri-strip^TM^ can help minimize scarring and improve scar quality. Scar-healing creams are products that claim to improve the appearance and texture of scars by moisturizing, softening, smoothing or fading them. There are many types of scar-healing creams, such as silicone-based creams, vitamin E creams, creams with natural ingredients, etc. However, there is not much medical evidence to support the effectiveness of these creams for scar healing. Some studies have shown that silicone-based creams may help to reduce the redness and itching of scars, but other studies have found no significant difference between the creams and a placebo. Vitamin E cream is sometimes recommended for scar management, but it can cause skin irritation or allergic reactions in some people. Creams with natural ingredients may have anti-inflammatory or antioxidant properties, but they may also contain allergens or irritants that can make scars worse [6].

For this reason, various strategies have been developed to treat postoperative scarring, including the early use of silicone gel/film, steroid injection therapy, laser treatment and surgical resection [7,8,9].

However, scars at the surgical site vary in shape, size and depth from person to person, making effective treatment difficult.

Scarring is also known to be influenced by individual sensitivity, race and genetic predisposition. In addition, the formation of scarring at the surgical site is considered difficult to predict and therefore difficult to prevent. The scarring process is broadly divided into three stages: inflammation, proliferation and maturation [10]. First, the inflammatory stage attracts various cells, including fibroblasts and vascular endothelial cells, which form blood clots and secrete various cytokines [11]. Second, in the proliferation stage, neovascularization provides sufficient blood flow to the wound and activates the proliferation of several cells, including fibroblasts [12]. Finally, during the maturation stage, there is continuous synthesis and degradation of collagen, which plays an important role in the remodeling of the wound substrate. If fibroblasts in the dermis proliferate abnormally and there is an imbalance in the continuous synthesis and degradation of collagen, a keloid or hypertrophic scar will result [13].

In 2018, the authors published a paper investigating the correlation between dermal thickness and scar formation. This study found that a thick dermis leads to poor scar formation and hypertrophic scarring [14]. In addition, it was predicted that the collagen distribution of the dermal layer may be a mechanism influencing scar formation, although further research is needed.

Collagen type I provides tissue strength and stability due to its characteristic amino acid sequence (glycine–proline–hydroxyproline), and collagen type III is associated with tissue elasticity [15]. Collagen type I is distributed in the reticular dermis, which accounts for most of the dermal thickness, and collagen type III is mainly distributed in the papillary dermis, a flat layer of the dermis [16]. Fetal wounds heal without scarring, and fetal fibroblasts are known to produce more type III collagen than those in older people [17]. The collagen type I/III ratio was increased in dilated cardiomyopathy [18]. Therefore, the authors hypothesized that the higher the dermal collagen type I/III ratio, the greater the likelihood of severe scarring. The aim of this study was to investigate the relationship between dermal collagen type I/III ratio and scarring.

## 2. Materials and Methods

### 2.1. Study Subjects

This single-institution, prospective, observational study was conducted on patients who visited the Department of General Surgery and Plastic and Reconstructive Surgery at Kosin University Gospel Hospital between September 2019 and September 2020 and who underwent round block technique (RBT) breast reconstruction immediately after breast-conserving surgery. Patients who underwent surgery and treatment at the expected incision, patients who received preoperative radiation therapy, patients who planned to receive postoperative radiation therapy, patients with <1 year follow-up after surgery and patients who underwent additional procedures at the surgical site, steroid users, patients with skin diseases, and immunocompromised patients were excluded from the study. A total of 78 patients were included in the study. All participants in this study gave written informed consent for their medical information to be stored in the database and for this information to be used for research purposes. This study was conducted after approval by the Clinical Trial Review Committee of Kosin University Gospel Hospital (approval number KUGH-2020-01-014).

### 2.2. Research Methods

#### 2.2.1. Tissue Sampling

All patients included in the study underwent immediate RBT breast reconstruction after breast-conserving surgery using the areola circumference incision method as described by Kim et al. in 2020 [19]. After drawing two concentric circles centered on the nipple, breast-conserving surgery was performed by making a semicircular incision along the arc where the breast cancer was located, and the breast was reconstructed through a glandular advancement flap around the site of the removed tumor. The semicircular area between the two concentric circles was removed and this skin tissue was used as the sample for the study. The rest of the area between the two concentric circles was de-epithelialized and then purse-string sutured (Figure 1).

#### 2.2.2. Immunohistochemistry

The collected skin sample was fixed in 4% formaldehyde for 12 h, dehydrated and embedded in olefin at 3 μm. Type I and type III collagen were detected in a limited area using polyclonal antibodies and the streptavidin–biotin–peroxidase complex (SABC) kit, and slides were exposed to 50 μL phosphate-buffered saline (PBS) to primary antibodies (mouse anti-human type I collagen or mouse anti-human type III collagen) diluted 1:100 in 50 g/L bovine serum antibody (BSA) at room temperature for 7 h. The next day, the slide was adapted at room temperature for 1 h, incubated with 50 μL biotinylated anti-mouse IgG for 1 h, incubated with SABC solution for 1 h, followed by incubation with 3,3′-diaminobenzidine for 10 min. Finally, the slide was washed with PBS (Figure 2).

#### 2.2.3. Measurement of Collagen Type I/III Ratio

Skin tissues were stained in picrosirius solution (0.1% solution of Sirius Red F3BA in saturated aqueous picric acid, pH 2) for 1 h for cross-polarization microscopy. Sections were washed in 0.01 N HCl for 2 min, dehydrated, cleared and mounted in synthetic resin. To analyze the collagen type I/III content and ratio, tissue samples were digitally scanned using an Axioscan 7 slide scanner and Zen slide scan digital image analysis software version 3.4.91 (Carl Zeiss GmbH, Jena, Germany). After selecting six regions of interest (50 × 50 pixels), the images were extracted through a masking process and converted to white pixels and black for the rest. By counting the white pixels using digital image analysis software (Image-Pro Plus, Media Cybernetics, Silver Spring, MD, USA), the collagen content by type was calculated, and, finally, the collagen type I/III ratio was analyzed (Figure 3).

#### 2.2.4. Post-Operative Management

All patients included in the study underwent closed suction drainage (Baro-Vac; Sewoon Medical, Cheonan, Republic of Korea). The authors removed the drain when the drainage volume became <10 mL/day, which was performed on average on postoperative day (POD) 5. No patient had a drain for more than 11 days, and no complications such as seroma occurred after drain removal. Then, the patient was discharged from the hospital, except in the case of a particularly large 24 h drainage volume. An intravenous antibiotic (ceftazidime, 2 g/day) was injected until the closed suction drain was removed. During hospitalization, the surgical site was dressed daily with a silicone foam dressing and antibiotic ointment, and after discharge, the dressing was applied in the same way every two days in the plastic surgery outpatient clinic. The surgical site suture was removed on POD 14. Clinical photographs were taken before surgery and at 1, 2 and 6 months and 1 year.

## 3. Evaluation Items

### 3.1. Baseline Characteristics

The patient’s age, BMI, lesion location and preoperative treatment were determined at the outpatient clinic visit before surgery, and the weight of the resected breast, cancer pathology, and the presence of postoperative radiotherapy were checked. The collagen type I/III ratio was determined based on the collagen type I and III contents obtained by the method described above.

### 3.2. Postoperative Scar Assessment

Photographs were taken with a wide-angle digital camera on the day before surgery and at 1, 2 and 6 months and 1 year after surgery. Based on standardized clinical photographs at 1 year post-operatively, each scar was scored using the Vancouver Scar Scale (VSS), which assesses pigmentation, pliability, scar height and vasculature. The scores for each of the four criteria were combined and graded by two independent plastic surgeons (Table 1). The lower the score (0–13 points), the better the scar, and the higher the score, the worse the condition (Figure 4).

## 4. Statistical Analysis

Statistical analysis was performed using SPSS ver. 18.0 (SPSS Inc., Chicago, IL, USA), and statistical significance was defined as *p*-values < 0.05.

Frequency analysis was performed on the baseline characteristics of study participants. Descriptive statistics are presented as medians with interquartile ranges or as numbers and percentages.Reliability analysis was performed to determine whether postoperative scan measurements were significantly consistent between two plastic surgeons.Pearson correlation analysis was performed to understand the relationship between baseline patient characteristics (age, BMI, weight of resected breast tissue, pathology, location, previous treatment history), collagen type I, III content and collagen type I/III ratio.Multiple regression analysis was performed with VSS as the dependent variable to verify the effect of items with significant results on scar formation.

## 5. Results

### 5.1. Demographic and Clinical Characteristics

Of 100 patients, a total of 78 were included in the study, excluding 3 patients who underwent postoperative radiotherapy and 19 patients with <1 year follow-up. The mean age was 48.77 ± 9.40 years. The mean BMI was 23.76 ± 3.92 kg/m^2^. The mean weight of resected breast tissue was 25.79 ± 11.98 g. Among all patients, ductal carcinoma in situ (DCIS) was the most common type, found in 40 patients (51.3%), and invasive ductal carcinoma (IDC) was the second most common type, found in 36 patients (46.2%). The most common tumor location was upper lateral (UL) in 32 patients (41%), upper medial (UM) in 22 (28.2%), lower medial in 14 (17.9%) and lower lateral (LL) in 10 (12.8%). There were 15 patients (19.2%) with previous treatment history, of whom, 8 patients (10.2%) had received preoperative chemotherapy and 4 (5.1%) had received hormone therapy. The collagen type I content measured by immunofluorescence staining and digital imaging was 31.63 ± 18.22 μg/g and the collagen type III content was 11.41 ± 6.27 μg/g. The collagen type I/III ratio was 3.85 ± 3.72 (Figure 5) (Table 2).

### 5.2. Scar Evaluation

The mean VSS scores evaluated by two independent plastic surgeons were 1.92 ± 2.01 and 1.79 ± 1.89, respectively, showing high overall cosmetic results. Reliability analysis was performed to verify the internal consistency of the two raters. As a result, the intraclass correlation coefficient was high at 0.920, and the reliability of the two specialists’ scores was judged to be good (*p* < 0.001) (Table 3).

### 5.3. Analysis of Correlations between Variables

Pearson’s correlation analysis was performed to confirm the correlation between the main variables of this study: VSS, age, BMI, the weight of resected breast tissue, collagen type I content, collagen type III content and collagen type I/III ratio. As a result, the collagen type I/III ratio showed a significant positive correlation with collagen type I content (r = 0.455, *p* < 0.01) and a significant negative correlation with collagen type III content (r = −0.340, *p* < 0.05). VSS showed a significant positive correlation with collagen type content (r = 0.412, *p* < 0.05) and collagen type I/III ratio (r = 0.552, *p* < 0.01) and a significant negative correlation with collagen type III (r = −0.326, *p* < 0.05). In contrast, age, BMI and weight of resected breast tissue showed no significant correlation with other variables (Table 4).

### 5.4. Factors Influencing Mean VSS

Multiple linear regression analysis was performed to verify the effect of collagen type l content, collagen type III content and collagen type I/III ratio, which showed a significant correlation with the dependent variable VSS. As a result, the regression model was statistically significant (F = 3.255, *p* < 0.01), and the explanatory power of the regression model was approximately 38.7% (R^2^ = 0.387, adjR^2^ = 0.268). The Durbin–Watson statistic was 1.361, indicating that the residuals could be considered independent, and the variance infusion factor was also less than 10, indicating no multicollinearity. As a result of testing the significance of the regression coefficient, it was found that the collagen type I/III ratio (β = 0.415, *p* = 0.028) had a significant positive (+) effect on VSS. In other words, VSS increased as the collagen type I/III ratio increased. On the other hand, collagen type (β = 0.219, *p* = 0.189) and collagen type III (β = −0.167, *p* = 0.281) had no significant effect on VSS (Table 5).

## 6. Discussion

The purpose of this study was to determine the correlation between the collagen type I/III ratio and the incidence of scarring in patients who underwent immediate reconstruction with the round block technique (RBT) after breast-conserving surgery. This is an important area of study because scarring can greatly affect a patient’s quality of life and self-esteem. The results of this study showed that the higher the collagen type I/III ratio, the more visible scarring could occur, whereas the lower the collagen type I/III ratio, the less visible and higher-quality scarring could occur.

This appears to be similar to previous studies showing a correlation between collagen type I/III ratio and scarring. A study published in 2011 by Cheng et al. found that the collagen type I/III ratio was significantly higher in hypertrophic scars than in normal skin [20]. In addition, in a study of hypertrophic scarring in human skin samples and a mouse model, Cameron et al. reported that the more severe the scar development, the higher the collagen type I/III ratio [21].

According to a recent study on the mechanism of scar formation, the wound healing process undergoes an inflammatory phase by neutrophils and macrophages for several days, followed by skin epithelialization, angiogenesis and fiber formation for several weeks. This proliferative phase is followed by collagen remodeling of the extracellular matrix. This increases over months to years and allows the skin to heal [22,23,24,25]. Factors that influence wound healing include local ischemia, anemia, steroid use, malnutrition, smoking, foreign bodies, infection, radiation, diabetes and systemic diseases such as renal failure. The overproduction of collagen is one of the many causes of keloid and thickening scars [9,24,26,27]. Keloid fibroblasts have been reported to have high proliferation and induce the hyperplasia of collagen. In addition, keloid fibroblasts exhibit fibronectin biosynthesis levels approximately four times higher than fibroblasts present in normal scars or dermis. In addition, various cells increase the expression of TGF-β during the wound healing process, which induces chemotaxis, proliferation and collagen production to the wound area [28,29,30].

In a previous study, we announced that the greater the dermal thickness, the worse the cosmetic result of scarring, the negative effect [14]. Collagen type I is distributed throughout the dermal layer, mainly in the reticular dermis, and collagen type III is distributed only in the papillary dermis and subpapillary dermis [31]. Since the reticular dermis occupies most of the dermal layer, it can be inferred that the thicker the dermal layer, the higher the content ratio of collagen type I compared to that of collagen type III. Research on the mechanism itself needs to be further discussed, but studies have been conducted on the occurrence of scarless healing in fetuses with high collagen type III content [17]. Therefore, the authors of this study thought that the higher the ratio of collagen type I to collagen type III content, the more active scar formation will be.

There are a few other things to note in our results. First, the collagen type I/III ratio was measured using a quantitative method. The authors quantitatively analyzed the collagen content through a digital scanning process after staining the tissue, a method similar to that used in studies on the correlation between the collagen type I/III ratio and hernia (2014) and the relationship between the collagen type I/III ratio and hemorrhoidal disease (2018). Cheong W et al. measured the collagen ratio value of normal adult skin to be 2.71 ± 0.63 and E. Peters et al. measured it to be 7.2 (interquartile range, 6.8–7.7). In the current study, the collagen ratio value was measured to be 3.85 ± 3.72, which is in between the two values [20,32]. In this study, the relationship between the collagen type I/III ratio and scar formation was investigated using an image conversion program (Axioscan 7 slide scanner and Zen slide scan digital image analysis software version 3.4.91) based on open-source computer vision. This allowed the accurate quantitative measurement of the collagen type I/III ratio. The advantage of this method is that it provides a more precise and accurate measurement than the conventional qualitative method [32,33]. Second, this study used VSS, a widely accepted and validated tool for measuring scar affinity, to evaluate scars. VSS has been used in many studies because it is a reliable and useful tool for scar assessment [34,35,36]. Third, another advantage of this study is that the relationship between the collagen type I/III ratio and scar formation could be understood in more detail by using multiple linear regression analysis to identify the factors affecting scar formation. The use of multiple linear regression analysis allows for a better understanding of the complex relationships between different variables, which in this case is the relationship between the collagen type I/III ratio and scar formation. This is one of the merits of the study as it provides a more comprehensive understanding of the underlying mechanisms that contribute to scar formation. Overall, the study provides valuable insights into the relationship between the collagen type I/III ratio and scar formation and may have implications for the development of interventions that can improve the appearance of scars in patients with reduced quality of life and low self-esteem due to scars.

In our study, we focused on the relationship between collagen type I/III ratios and scar formation in breast cancer patients after surgery. However, it is also important to consider the potential influence of clinical characteristics and genetic factors on these collagen ratios and scar formation. Previous research has shown that certain genetic factors, such as single nucleotide polymorphisms in collagen genes, can influence collagen production and organization, leading to variability in scar quality between individuals [37,38].

In addition, clinical characteristics such as age, BMI and comorbidities may also influence collagen synthesis and degradation, ultimately affecting the healing process and scar quality [39]. Therefore, it would be valuable in future research to investigate the contribution of these factors to the collagen type I/III ratio and their effect on scar formation in breast cancer patients. A more complete understanding of these factors may help to develop personalized treatment approaches and improve patient outcomes.

Based on the results of this study, the authors believe that research into the genetic factors that influence the collagen type I/III ratio may help to develop a patient-specific scar prediction model for practical use. Genetic blood testing will allow clinicians to know the patient-specific collagen type I/III ratio before surgery and predict the degree of scarring based on this information, thereby expanding the choice of patients to determine incision or prevent scarring.

This study has several limitations. First, it was a single-center study with a relatively small sample size of 78 patients. This may limit the generalizability of the results. Second, because scars were evaluated by VSS using clinical photographs, clinical symptoms associated with scars, such as pain, itching and pulling, were not reflected. Third, this study evaluated scars based on clinical photographs during the first year of postoperative follow-up, and it is believed that a longer period of follow-up is needed to understand the long-term effect of collagen type I/III ratio on scar formation. Fourth, all participants in this study were patients who underwent oncoplastic breast-conserving surgery for breast cancer. Therefore, the results of this study may be difficult to generalize to patients who have undergone surgery in other areas or who are not cancer patients. Finally, three patients who received radiotherapy after surgery were excluded, and there is a possibility that this may have introduced selection bias.

## 7. Conclusions

In conclusion, this study analyzed skin tissue obtained from patients who underwent immediate reconstruction with the round block technique (RBT) after breast conservation surgery. The result of the VSS evaluation of scars by clinical photographs in the first year after surgery showed that the degree of scarring was related to the collagen type I/III ratio of the dermal layer, and the lower the collagen type I/III ratio, the better the cosmetic result of the scar. Based on these results, it is expected that in the future, it will be possible to predict the degree of scarring in patients before surgery, which will help determine the incision area for surgery or prevent scarring.

## Figures and Tables

**Figure 1 biomedicines-11-01089-f001:**
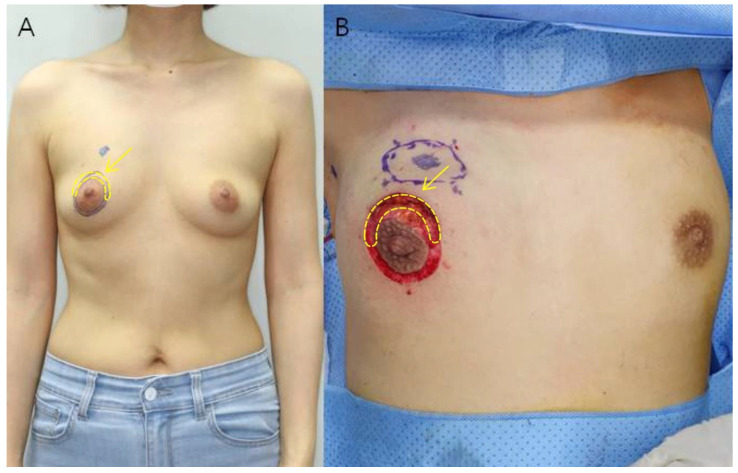
(**A**) Preoperative photograph of a 46-year-old woman. The tumor is located in the upper inner quadrant of the right breast. After drawing two concentric circles centered on the nipple, a semicircular incision is made along the arc of the breast cancer site to perform breast conservation surgery. (**B**) Intraoperative photograph after tumor resection. Deep dissection is performed between the two concentric circles and the semicircular part of the breast cancer site is excised. Excess skin was excised to obtain the specimen (yellow arrow).

**Figure 2 biomedicines-11-01089-f002:**
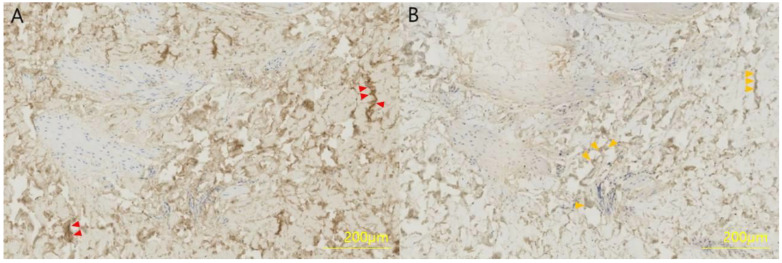
Immunohistochemistry of a full-thickness skin specimen from a 46-year-old woman. (**A**) Collagen type I and (**B**) collagen type III were formalin-fixed, paraffin-embedded and stained with a polyclonal antibody at a dilution of 1:100 for 4 h. Collagen type I is stained brown (red arrows) and collagen type III is stained light brown (orange arrows).

**Figure 3 biomedicines-11-01089-f003:**
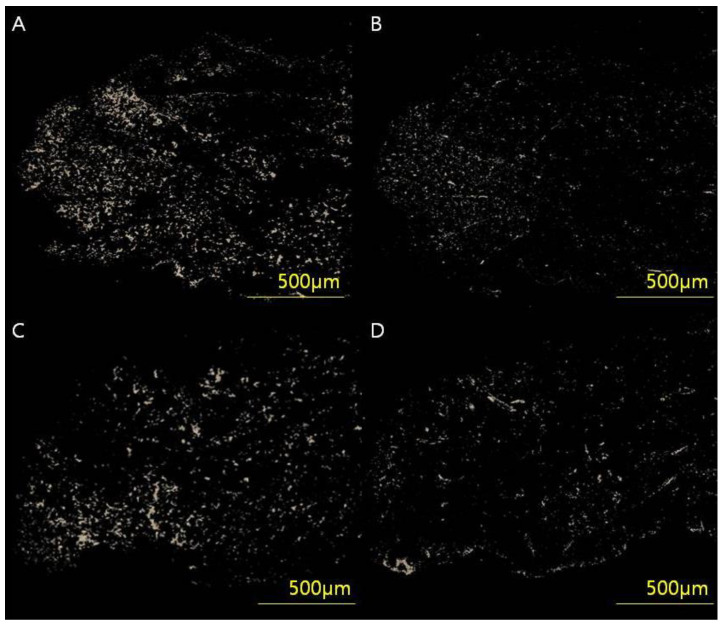
(**A**,**B**) Full-thickness skin samples from a 54-year-old woman (VSS 4) who underwent breast-conserving surgery through a periareolar incision were acquired and analyzed using an Axioscan 7 slide scanner and Zen Slide Scan digital image analysis software version 3.4.91 (Carl Zeiss GmbH, Jena, Germany). (**C**,**D**) Full-thickness skin samples from a 53-year-old woman (VSS 0). Six random areas (50 × 50 pixels) were captured from each image, and the quantitative areas of collagen type I/III were analyzed using digital image analysis software (Image-Pro Plus, Media Cybernetics, Silver Spring, MD, USA) and custom software to obtain collagen type I/III content and ratio. The analyzed values of (**A**) collagen type l content, (**B**) collagen type III content and type I/III ratio were 34.78 μg/g, 11.09 μg/g and 3.13, respectively. The analyzed values of (**C**) collagen type I content, (**D**) collagen type III content and type I/III ratio were 13.56 μg/g, 14.96 μg/g and 0.90, respectively.

**Figure 4 biomedicines-11-01089-f004:**
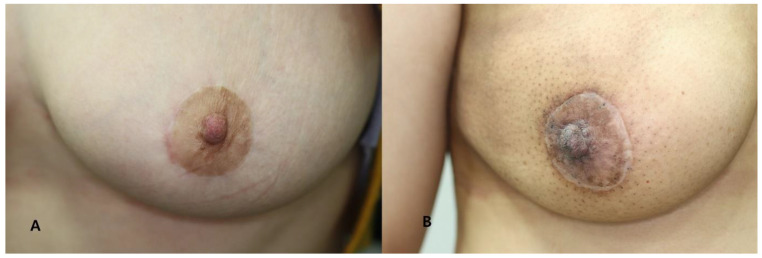
Clinical photographs taken at 12 months postoperatively. (**A**) A 39-year-old female patient assigned to the good scar group (VSS score: 0). (**B**) A 47-year-old female patient assigned to the poor scar group (VSS score: 4).

**Figure 5 biomedicines-11-01089-f005:**
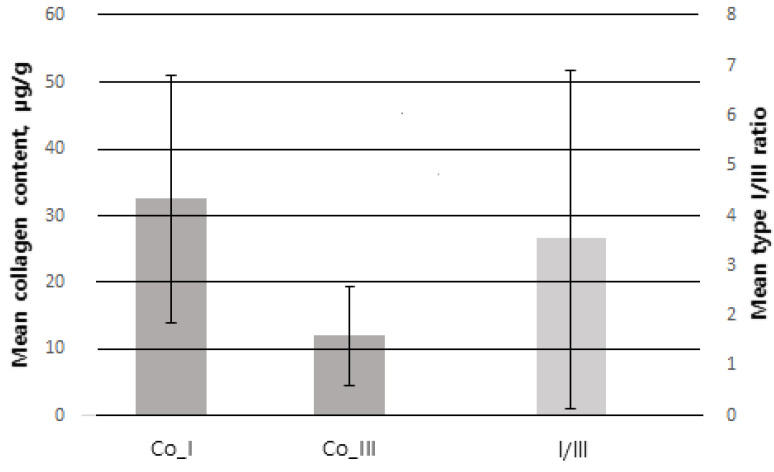
Error bar of SD (standard deviation). Co_I collagen type I (SD = 18.22), Co_III = collagen type III (SD = 6.27) follow the left *Y*-axis. I/III = collagen type I/III ratio (SD = 3.72) follow the right *Y*-axis.

**Table 1 biomedicines-11-01089-t001:** The Vancouver Scar Scale (VSS).

Scar Characteristic	Score
Vascularity	
Normal	0
Pink	1
Red	2
Purple	3
Pigmentation Normal	0
Hypopigmentation	1
Hyperpigmentation	2
Pliability	
Normal	0
Supple	1
Yielding	2
Firm	3
Ropes	4
Contracture	5
Height (mm)	
Flat	0
<2	1
2~5	2
>5	3
Total score	13

**Table 2 biomedicines-11-01089-t002:** Baseline patient characteristics.

Characteristic	Value (%)
Total no. of patients	78
Mean age (year)	48.77 ± 9.40
Mean BMI (kg/m^2^)	23.76 ± 3.92
Weight of resected breast tissue (g)	25.79 ± 11.98
Tumor type	
DCIS	40 (51.3)
LCIS	2 (2.6)
IDC	36 (46.2)
ILC	0 (0)
Tumor Location	
UL	32 (41)
UM	22 (28.2)
LL	10 (12.8)
LM	14 (17.9)
Previous treatment history	
Yes	12 (15.3)
Preoperative chemotherapy	8 (10.2)
Preoperative hormonal therapy	4 (5.1)
No	63 (80.7)
Mean collagen content	
Collagen Type I (µg/g)	31.63 ± 18.22
Collagen Type III (µg/g)	11.41 ± 6.27
Collagen type I/III ratio	3.85 ± 3.72

Values are expressed as mean ± SD (Standard Deviation) or number (%). BMI = body mass index, DCIS = ductal carcinoma in situ, LCIS = lobular carcinoma in situ, IDC = invasive ductal carcinoma, ILC = invasive lobular carcinoma, UL = upper lateral, UM = upper medial, LL = lower lateral, LM = lower medial.

**Table 3 biomedicines-11-01089-t003:** Reliability analysis between 2 independent plastic surgeons.

	Mean VSS	Interclass Correlation	*p*-Value
Doctor A	1.92 ± 2.01	0.920	<0.001 *
Doctor B	1.79 ± 1.89		

* = Statistically significant. Values are expressed as mean ± SD (standard deviation).

**Table 4 biomedicines-11-01089-t004:** Analysis of the correlation between the main variables.

Variables	1	2	3	4	5	6	7
1. Age	1						
2. BMI	0.237	1					
3. WRBT	0.009	0.112	1				
4. Co_I	0.112	−0.030	−0.065	1			
5. Co_III	0.075	−0.022	−0.068	−0.070	1		
6. I/III	0.172	0.236	0.139	0.455 **	−0.340 *	1	
7. VSS	−0.013	0.118	−0.044	0.412 *	−0.326 *	0.552 **	1

* = *p* < 0.05, ** = *p* < 0.01, BMI = body mass index, WRBT = weight of resected breast tissue, Co_I = collagen type I content, Co_III = collagen type III content, I/III = collagen type I/III ratio, VSS = Vancouver Scar Scale

**Table 5 biomedicines-11-01089-t005:** Multiple regression analysis of Vancouver Scar Scale (VSS) and correlating factors.

Dependent Variable	Independent Variable	B	S.E.	β	*t*	*p*-Value	VIF
VSS	C	1.594	1.191		1.338	0.191	
	Co_I	0.015	0.011	0.219	1.344	0.189	1.338
	Co_III	−0.024	0.022	−0.167	−1.097	0.281	1.169
	I/III	0.086	0.037	0.415	2.312	0.028	1.625
F = 3.255 (*p* < 0.01), R^2^ = 0.387, _adj_R^2^ = 0.268, D-W = 1.361							

B = unstandardized coefficient. S.E. = standard error. β = standard coefficient. VIF = variance inflation factor. C = constant. R^2^ = R-square., _adj_R^2^ = adjusted R-square. D-W = Durbin–Watson test.

## Data Availability

The datasets generated during and/or analyzed during the current study are available from the corresponding author on reasonable request.

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
