# Peer review of "Correlation between Collagen Type I/III Ratio and Scar Formation in Patients Undergoing Immediate Reconstruction with the Round Block Technique after Breast-Conserving Surgery"

_biomedicines, 2023, doi:10.3390/biomedicines11041089_

Round 1

Reviewer 1 Report

This manuscript aims to reveal the relationship between collagen type I/II ratio and scar formation in breast cancer patients with the surgery. Also, the authors indicate that such ratio can be used to predict the quality of scar formation. Since scarring negatively influences the life quality of patients, scarring prediction becomes an interesting subject to study. Despite the authors using patient samples, the experimental design in this manuscript needs improvement to reduce confusion and experimental results cannot support the conclusions very well. Here are some comments:

1) Many results lack the control group (see Figure 2, Figure 3, and Figure 5). This means we cannot compare the collagen content in these groups with control groups and thus cannot tell whether the collagen shown in these figures is relatively high or low.

2) Microscopic photos lack the scale bar (see Figure 2 and Figure 3) and need arrows to point out collagen content such as in Figure 2.

3) Line 129-139. The collagen type I/II ratio measured by analyzing microscopic images is tedious and inaccurate. Due to the limitation of the microscope in counting cells with specific properties, people invented flow cytometry, which can easily measure 10,000 cells in one minute. If possible, I recommend authors to use the flow cytometer to do this experiment again.

4) Authors should discuss the role of clinical characteristics in determining collagen type I/II ratio or influencing scarring formation. Because patients with different disease characteristics could be the result of genetic factors. Indeed, the authors mentioned that genetic factors likely affect the collagen type I/II ratio.

Author Response

Thank you for reviewer's insightful comments

Reviewer 2 Report

General comment: The topic is of great interest, prospective on 78 patients undergoing conservative surgery for breast cancer. The authors show their experience on the correlation between collagen for scar formation after round block technique. Many biological and mechanical factors are involved in scar formation. The authors suggest a collagen assessment of scars.

Some suggestions:

1) improve introduction on the effects of mechanical forces that cause scar alteration, especially in areas of skin where the forces are stretched. What strategies exist, immediate post surgery? the use of creams other than silicone? to help the formation of a good scar

2) clarify the inclusion criteria: were patients with skin diseases excluded? Patients who are immunocompromised or treated with steroids?

3) was volume in the drain considered normal to remove it? Have there been any cases of excessive whey production or collection?

I would advise authors to revisit their literature search and at least add these works:

Oswaldo JIE et al. Effects of mechanical forces on the formation of cutaneous wounds during skin expansion and emerging therapies for wound healing and scar prevention. Saudi Med J. 2023 Jan;44(1):106-109. doi: 10.15537/smj.2023.44.1.20220556.

Aziz J, Ahmad MF, Rahman MT, Yahya NA, Czernuszka J, Radzi Z. AFM analysis of collagen fibrils in expanded scalp tissue after anisotropic tissue expansion. Int J Biol Macromol 2018; 107:1030-1038.

However, we must consider the effort of the authors worthy of attention

- Clarify all abbreviations used in the text

-The images are of good quality and standardized.

Author Response

Thank you for reviewer's insightful suggestions.

Round 2

Reviewer 1 Report

Thanks for the improvement and response!